# Measuring Urban Greenspace Distribution Equity: The Importance of Appropriate Methodological Approaches

**Meghann Mears *** and **Paul Brindley**

Department of Landscape Architecture, University of Sheffield, Floor 13, The Arts Tower, Western Bank, Sheffield S10 2TN, UK; p.brindley@sheffield.ac.uk
* Correspondence: meghann.mears@sheffield.ac.uk

**Abstract:** Urban greenspace can provide physical and mental health benefits to residents, potentially reducing health inequalities associated with socioeconomic deprivation. The distribution of urban greenspace is an important social justice issue, and consequently is increasingly studied. However, there is little consistency between studies in terms of methods and definitions. There is no consensus on what comprises the most appropriate geographic units of analysis or how to capture residents' experience of their neighbourhood, leading to the possibility of bias. Several complementary aspects of distribution equity have been defined, yet few studies investigate more than one of these. There are also alternative methods for measuring each aspect of distribution. All of these can lead to conflicting conclusions, which we demonstrate by calculating three aspects of equity for two units of aggregation and three neighbourhood sizes for a single study area. We make several methodological recommendations, including taking steps to capture the relevant neighbourhood as experienced by residents accurately as possible, and suggest that using small-area aggregations may not result in unacceptable levels of information loss. However, a consideration of the local context is critical both in interpreting individual studies and understanding differing results.

**Keywords:** urban greenspace; equity mapping; inequality; geographic information systems; modifiable areal unit problem; unknown geographic context problem

## 1. Introduction

There is mounting evidence for the positive effect of exposure to greenspace on human health and well-being [1–5]. These benefits derive from a variety of mechanisms including reduction of stress and psychological restoration, promotion of social cohesion, provision of space for physical activity, reduction of exposure to harmful environmental conditions (heat, noise and air pollution), and immune system modulation [1,2,5]. The improved health outcomes are equally varied, including reductions in cardiovascular disease and all-cause mortality, and increased birth weight and physical activity levels [1]. Moreover, there is evidence that greenspace has the potential to reduce health inequalities associated with socioeconomic deprivation [6–10]. Given that deprived groups have the greatest need of goods and services that improve health, failure to provide adequate greenspaces for more deprived groups is considered an environmental inequity (as opposed to an inequality, whereby everybody would receive the same resources, regardless of need) [11,12]. Access to greenspace has therefore become an important environmental justice issue [13–16].

While studies investigating greenspace distribution equity are increasingly common, there is considerable diversity in methods and definitions used to quantify equity. Some of this variation arises from limitations of the available data. Census data relating to socioeconomic deprivation and health is

usually available only at aggregated neighbourhood scales in order to preserve anonymity. In England and Wales, for example, the smallest unit at which data is made available is the Output Area (OA), which has an average population of 309 [17]. Suitable data on actual greenspace usage or exposure data are also rarely available [1]. GIS software, remotely sensed data and land cover/use databases provide methods for estimating exposure [1,18], but due to the computational intensity of certain GIS procedures, it may not be feasible to perform household-level calculations for large areas even where deprivation/health data is available at this scale.

These issues of scale and aggregation relate to the well-known modifiable areal unit problem (MAUP): there are many ways in which spatial data can be aggregated, and both the scale and basis for aggregation have the potential to strongly affect observed spatial patterns and thus relationships with other variables [19,20]. Also relevant is the ecological fallacy, in which patterns observed in aggregated data do not hold in analysis of individuals [20,21].

Another cause of variation relates to the uncertain geographic context problem (UGCoP), which describes the difficulty of knowing the "true causally relevant" spatial context in terms of how individuals use and experience their environment [22,23]. This is a relevant issue for analysis of greenspace exposure, which would ideally record which greenspaces people experience or travel near regularly in their day to day movements in order to minimise contextual assumptions [23]. Such data is rarely available for large samples; although advances in technology such as smartphone GPS tracking show promise in this area [1,8,18,24,25].

Given these limitations and implicit assumptions, there is little consensus as to how to estimate greenspace distribution in ways that are most relevant and effective for identification of inequities [14,26]. There are three broad components of distribution, of which few studies (except [26,27]) consider all three. *Accessibility* is often operationalised as distance to greenspace or proportion of population within a given distance of greenspace [13,15,26,28–36]. *Provision*, or amount of greenspace cover, is perhaps the simplest and easiest to quantify, and consequently is the most commonly used type of metric, especially in studies of large areas [10,26,29,32,36–40]. The least commonly used concept, *population pressure*, is the potential for crowding within greenspaces, and is calculated using an assumption such as everyone visiting their nearest greenspace simultaneously [13,15,26].

Within each of these categories, a range of approaches exists to associate the greenspace measures with the local population or neighbourhood. Some studies simply use the boundaries of census tracts, for example, calculating the tracts' percentage greenspace cover [10,40]. More common in studies of social equity (as opposed to health) is the use of a buffer around a census tract boundary or population centroid, or around individual households. With this method, a range of buffer distances are used. UK guidance suggests that everyone should live within 300 m (0.18 miles), equivalent to five minutes' walk, of a greenspace [3]. This is also the distance recommended by a review paper as an indicator of greenspace accessibility [41]. While some studies use this buffer distance [42–45], others use distances ranging between 400 m (0.25 miles) and 3200 m (2 miles) [13,14,26,28,29,32,33,38,39]. Yet other studies avoid assuming an appropriate neighbourhood size by simply using distance to nearest greenspace [13,26,29–31,34–36]. While there may be a sharp drop-off in greenspace visitation frequency with distance [46–48], a review of studies of the relationship between greenspace provision and physical health found effects of provision up to 2 km away [8], suggesting that more distant greenspaces also provide benefits.

There are also two approaches to constructing buffers and measuring distances: using straight-line, also called as-the-crow-flies distance [13,15,29,33,35,37,38]; and using network distance, which uses a GIS representation of the transport network to estimate real-world travel distances [26,28,30–32,34,36,39]. Under certain scenarios, these two approaches can lead to quite different estimates of the area accessible to residents (Figure 1).

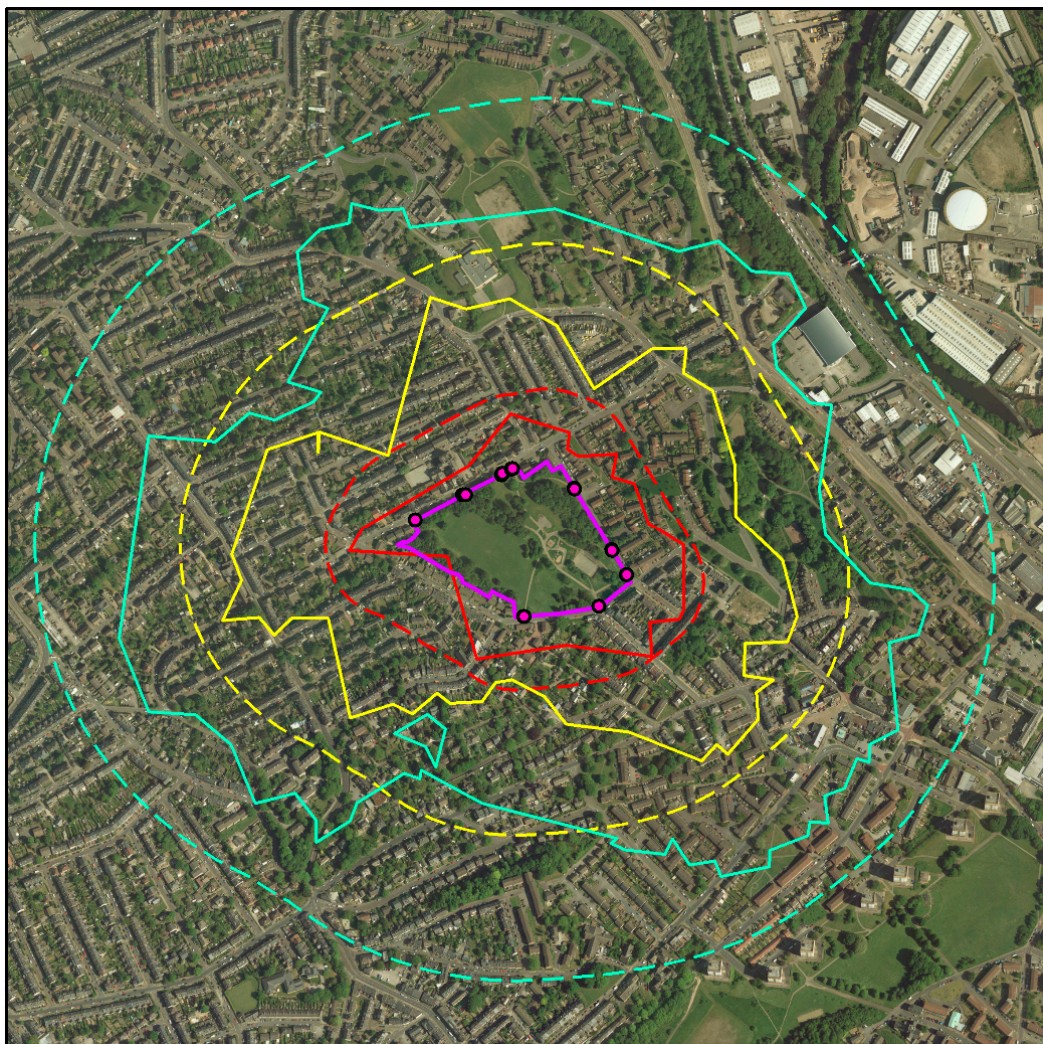

**Figure 1.** Comparison of areas within network buffers (solid lines) and straight-line buffers (dotted lines) of access points for a greenspace (purple points and boundary), at 100 m (red), 300 m (yellow) and 500 m (blue).

Given this variety in methodological approach, there is unsurprisingly little consistency in the results of studies. Many find that less socioeconomically deprived areas have greater opportunities for greenspace exposure [13,15,26,28,29,31,32,34,36–39]. However, others find no relationship [26,29,32], or even the opposite relationship [13,15,26,30,32–36,39]. Indeed, studies including multiple definitions of greenspace availability or exposure commonly find conflicting results for the same study area [15,26,29,32,34,36,39,49]. Furthermore, there appears to be no pattern in results according to methodological details or study area location [13,15,26,28–39]. Some studies using identical methods have found different results in different study areas [18]. This latter finding highlights both the importance of considering local contextual factors in selecting appropriate methods, such as historical factors determining where greenspaces are located and variation in their maintenance, and the demographic features of neighbourhoods [13,15,18,27]; and the limitations of indirect approaches to measuring exposure, arising from the MAUP and UGCoP issues described previously and the difficulty of collecting adequate data on actual greenspace usage.

In this study, we demonstrate the consequences of several methodological decisions using the case study city of Sheffield, UK. We investigate the question of how greenspace availability varies with socioeconomic deprivation using the accessibility, provision and population pressure concepts. These are analysed at both household level and a small area census geography, using six different approaches to estimating a contextually relevant neighbourhood (network and straight-line buffers,

both at three different distances). We use this example to illustrate the importance of making sound methodological choices in terms of both general approaches and the local context.

## 2. Materials and Methods

### 2.1. Study Area

Sheffield, UK (53°23′ N, 1°28′ W), is an inland city with a population of approximately half a million and administrative boundaries covering an area of 368 km². Sheffield is unusual in that there is a large expanse of sparsely populated moorland within its borders, with the vast majority of the population living in the urbanised east. A gradient of socioeconomic deprivation developed during the industrial revolution, with working class families living near the centres of industry along the rivers to the east of the urbanised area. Wealthier citizens lived further west, upwind and at higher elevation to avoid high levels of pollution. This pattern is still seen in income and health deprivation today [50].

Sheffield is a relatively green city, including a large number of public parks and other greenspaces. The location of many of Sheffield's parks and greenspaces was influenced by the socioeconomic gradient: public health measures to improve the health of the urban working class in the mid-nineteenth century included the establishment of parks in working class neighbourhoods [51]. Some other parks were initially established on the urban outskirts, and have remained as the city grew around them. Other greenspaces, often less intensively managed or seminatural, are situated due to local geographic features, such as on steep hills or banks unsuitable for development, or green corridors along canals.

### 2.2. Data

#### 2.2.1. Units of Analysis

We used two separate spatial units of analysis in order to analyse the effects of aggregation. The first was individual households, for which residential address points were identified from Ordnance Survey (OS—Great Britain's national mapping agency) AddressBase Plus. 541 address points were excluded from all analyses on the basis of being located within greenspace polygons. In the majority of cases, this was due to sites having been sold for development in the time between the greenspace assessment (2008) and address point data (2017; older data were not available), although a minority were due to minor digitisation accuracies or due to genuine residences, e.g., club houses.

The second unit of analysis was Output Areas (OAs). OAs are the smallest census geography, with an average population of 309. OA boundaries are drawn to represent socially homogeneous areas in terms of dwelling type and household tenure [52], so can be considered a type of localised neighbourhood.

Given our focus on urban areas, we only included OAs classed as urban in the 2011 Rural-Urban Classification, and only households within those OAs. This includes 50.3% of the total area within the administrative boundary, but 96.5% of the households. Including the same areas ensured comparability of household- and OA-level analyses. The number of units included at each scale are: OAs n = 1785 (out of 1817); address points n = 243638 (out of 252023). A map of included OAs, showing the location of households and greenspaces, is shown in Figure 2.

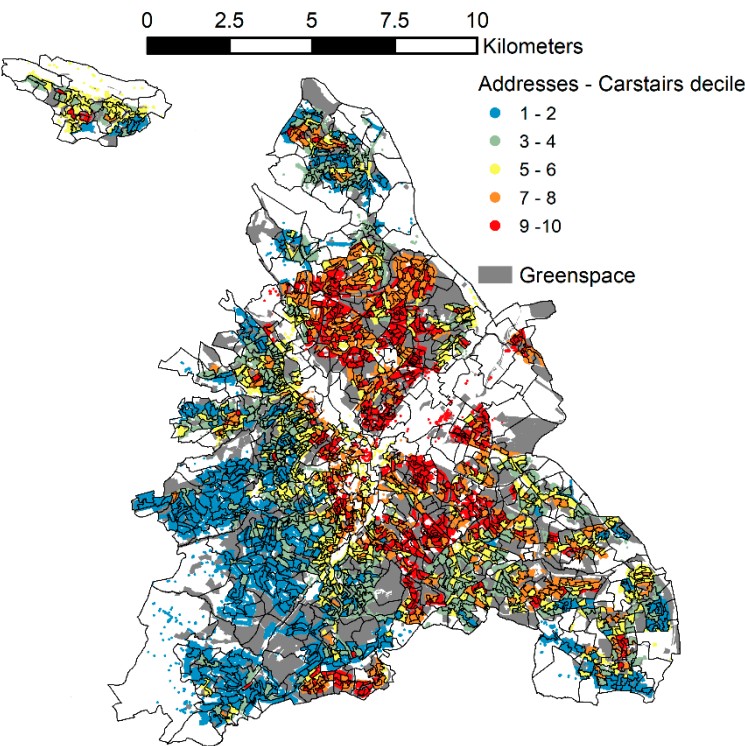

**Figure 2.** The urbanised Output Areas (OAs) of Sheffield (classed as 'urban'), with locations of households (coloured according to Carstairs deprivation index decile) and greenspaces.

### 2.2.2. Area Deprivation

As the measure of socioeconomic deprivation, Carstairs Deprivation Index [53] was calculated at the OA scale using 2011 census data. We used this index instead of the more commonly used Index of Multiple Deprivation (IMD) because the latter is only available at larger geographies; however, the correlation between IMD and Carstairs is high (0.96 at Lower Super Output Area scale).

For the household-level analysis, the OA Carstairs index was assigned to each individual household. For the purposes of statistical testing, we divided households and OAs into deciles of deprivation. Decile counts are slightly uneven for the household-level analysis due to assignment of a single value to all households within an OA.

### 2.2.3. Greenspace Data

Greenspace data were supplied by Sheffield City Council, and included "accessible open spaces, sports and recreation provision" [54] identified as part of the Council's 2008 PPG17 (Planning Policy Guidance 17: Planning for open space, sport and recreation) assessment. This comprised GIS polygon data, accompanied by attributes including type, size, selected amenities and assessments of greenspace quality. We excluded the six primarily hard-surfaced sites, alongside three sites for which we could not identify public access points (see below), leaving a total of 936 sites (Figure 2).

We mapped access points for each site. These were preferentially located from pre-existing data sources including Sheffield City Council Parks and Countryside Service data; OS Greenspace (Academic version); and a previous research project. Where these sources proved inadequate, we located further points by intersecting greenspace boundaries with transport network lines (described below), or visually through use of aerial photography and Google StreetView.

2.2.4. Transport Network

We built a network dataset using ESRI ArcGIS 10.1 to enable calculation of network distances. This combined OS Integrated Transport Network (roads and urban paths layers) and OpenStreetMap data (lines layer, only lines with the 'highway' attribute set). Each dataset included some paths that the other did not. Classes of road not accessible to pedestrians were excluded (motorways, motorway links, racetracks, roads under construction). The layers were snapped to minimise duplication. Additional paths were added to the dataset using aerial photography where access points were not within 10 m of the existing transport network. We additionally checked that greenspaces' internal paths were adequately mapped and that there were no significant isolated network segments. Network lines were manually added to correct issues where necessary.

*2.3. Accessibility Measure*

As network analysis is a computationally intensive process, it was not possible to compute network distances to greenspaces for each household individually. We therefore used ArcGIS Generate Service Areas tool to identify areas within 100 m, 300 m and 500 m of access points for each greenspace. A spatial select query was used to identify whether households were within the specified distance of each greenspace. Where output was required for OAs instead of households, we used population centroids as a single representative point for each area. The calculation of this accessibility measure was automated using Python and the ArcGIS module ArcPy.

We noted that the Generate Service Areas tool 'detailed polygons' option resulted in small holes in the output polygons. In some cases, the holes incorrectly coincided with address points (because address points do not always lie adjacent to roads). These holes were removed using the Eliminate Polygon Parts tool.

In contrast, straight-line buffer accessibility measures were generated from buffers around greenspace boundaries for the same distance bands. The greenspace was considered accessible to households and OA population centroids if they fell within the buffer.

For OAs, we additionally assessed areal coverage accessibility, i.e., whether any access points fell within OA boundaries.

*2.4. Provision Measure*

We defined greenspace provision as the total area of greenspaces with at least one access point within the specified distance of each address point or population centroid. The whole area of greenspaces with an access point within the distance was included, rather than just the area within that distance, as our distance bands were determined considering how far people will travel *to* greenspaces, rather than *within* them.

Provision was assessed at the same distance buffers and using the same buffer construction methods (network, straight-line) as for accessibility. For the areal coverage provision measure, we calculated the area of the intersect between OAs and greenspaces.

*2.5. Population Pressure Measure*

The population pressure measure was calculated using the assumption that all households visit a nearby greenspace simultaneously. Due to unavailability of usage data, we assumed equal likelihood of visitation to all sites within the distance band. First we calculated the population pressure for each greenspace as the number of households (or OA population centroids) within its service area, weighted according to the number of service areas in which that household fell (so that a household in the service area of only one site would contribute 1 to its total population, while a household in three service areas would contribute 1/3 to each site); and divided this by the site's area. To obtain the household- or OA-level population pressure, we then took the area-weighted mean population pressure of all sites in the provision for that household/OA.

This was achieved in GIS by taking the intersect of the service areas and address points/OA population centroids, and cross-tabulating by address to count the service areas for each address/OA point and calculate its weighting. We used a spatial join to sum the weighted points within service areas. Subsequently we joined this greenspace-level population back to address/OA points. This number was used as the numerator to calculate population pressure, while the denominator was the total area of greenspaces in the address/OA point's provision.

Population pressure was assessed at the same distance buffers and using the same buffer constructions methods as for the other measures. The areal coverage population pressure was calculated as the number of address points divided by the area of greenspace within the OA.

*2.6. Statistical Analysis*

Statistical analysis of accessibility comprised a binomial ANOVA testing whether accessibility varied between deciles of deprivation. Each household/OA was represented by either a 1 (has access) or a 0 (does not have access). This analysis was carried out in R v3.5.1 by calculating a type-III ANOVA table (function 'Anova' in library car) for a generalised linear model (function 'glm') using the binomial family [55,56]. Model diagnostics indicated that the model assumptions (residual distributions and homogeneity of variance) were met in all cases. Post-hoc Tukey multiple comparison tests were performed where *p*-values indicated a significant ANOVA test at $\alpha = 0.05$. Multiple comparison tests were performed using the emmeans library, which uses estimated marginal means to account for unbalanced design, to test for differences between each pair of deciles using a Tukey correction [57].

Provision and population pressure were analysed statistically using two-stage (hurdle) models, which were required due to the sometimes large number of zeroes arising from points not within any service areas. The first stage was therefore identical to the accessibility analysis (binomial ANOVA), while the second included only points within at least one service area to analyse inter-decile variability. The second stage comprised an ANOVA with post-hoc Tukey multiple comparison tests based on estimated marginal means. Second stage ANOVAs were again carried out using type-III ANOVA tables calculated for generalised linear models, but in this case using the gaussian family. Again, model assumptions were met in all cases. Multiple comparison tests used the same method as for binomial ANOVAs.

## 3. Results

*3.1. Accessibility*

Results of binomial ANOVAs of variation in accessibility by Carstairs decile are shown in Table 1. Decile means and results of post-hoc multiple difference tests are shown in Figure 3, except for OA-scale areal coverage which is shown in Figure 4a. Although the proportion of deviance (a measure of goodness of fit) explained by Carstairs decile is low (mean 6%; range 2–16%), all except one ANOVA are statistically significant at $\alpha = 0.05$, and there are significant multiple differences in all except two cases.

As expected, accessibility increases with distance considered. All models using a 100 m buffer (Figure 3a,d,g,j) show a generally increasing trend of accessibility with deprivation, i.e., higher proportions of more deprived households are within 100 m of a greenspace access point. This can also be observed for the 300 m models using network analysis (Figure 3b,h). However, at greater distances nearly all households/OAs have access, so this trend is less apparent or not in evidence at all (Figure 3c,e,f,i,k,l). This pattern is only slightly apparent in the areal coverage analysis, although here the least deprived decile has higher levels of accessibility than other low-deprivation deciles (Figure 4a).

There is little difference between the results for household-scale and OA-scale analyses (e.g., compare Figure 3a,g, or Figure 3d,j). At 100 m, the overall proportion of households with access is 12% higher than the proportion of OAs with access, but at other distances the overall proportions are within 1% of each other.

In contrast, using buffer analysis overestimates accessibility compared to network analysis, especially at smaller distances (example shown in Figure 1). At 100 m, buffer analysis estimates of the overall proportion of households and OAs with access respectively are 248% and 276% of the estimates using network analysis. At 300 m, the figures are 128% and 129%, respectively; although at 500 m, due to most points being within 500 m of an access point using network analysis, the differences are negligible. Overestimates are particularly large for less deprived deciles, where levels of access are lower.

The results of the areal coverage analysis are most similar to the buffer analyses at 100 m (Figure 4a, Figure 3d,j), although the areal coverage analysis indicates a weaker relationship.

**Table 1.** Results of ANOVAs testing differences in greenspace accessibility, and provision and population pressure for households/OAs with access to at least one greenspace, by decile of Carstairs Deprivation Index. Deviance explained shown as a percentage. Statistically significant *p*-values are shown in bold. Proportion of households/OAs with access shown to indicate loss of statistical power in some provision and population pressure analyses (total households n = 243,638; total Output Areas n = 1785).

| Model | | % with Access | Accessibility | | Provision | | Population Pressure | |
|---|---|---|---|---|---|---|---|---|
| | | | *Deviance Explained* | *p* | *Deviance Explained* | *p* | *Deviance Explained* | *p* |
| **Households** | | | | | | | | |
| Network | 100 m | 21 | 4 | **<0.001** | 2 | **<0.001** | 8 | **<0.001** |
| | 300 m | 74 | 5 | **<0.001** | 2 | **<0.001** | 4 | **<0.001** |
| | 500 m | 95 | 5 | **<0.001** | 2 | **<0.001** | 3 | **<0.001** |
| Buffer | 100 m | 51 | 2 | **<0.001** | 2 | **<0.001** | 6 | **<0.001** |
| | 300 m | 95 | 3 | **<0.001** | 1 | **<0.001** | 3 | **<0.001** |
| | 500 m | 100 | 12 | **<0.001** | 1 | **<0.001** | 3 | **<0.001** |
| **Output Areas** | | | | | | | | |
| Network | 100 m | 18 | 5 | **<0.001** | 5 | **0.041** | 5 | **0.041** |
| | 300 m | 74 | 6 | **<0.001** | 1 | 0.076 | 2 | **<0.001** |
| | 500 m | 95 | 7 | **<0.001** | 2 | **<0.001** | 2 | **<0.001** |
| Buffer | 100 m | 51 | 2 | **<0.001** | 3 | **0.001** | 4 | **<0.001** |
| | 300 m | 95 | 4 | **<0.001** | 1 | **0.026** | 2 | **<0.001** |
| | 500 m | 100 | 16 | 0.613 | 1 | **0.029** | 3 | **<0.001** |
| Areal | | 55 | 2 | **<0.001** | 3 | **0.003** | 3 | **0.002** |

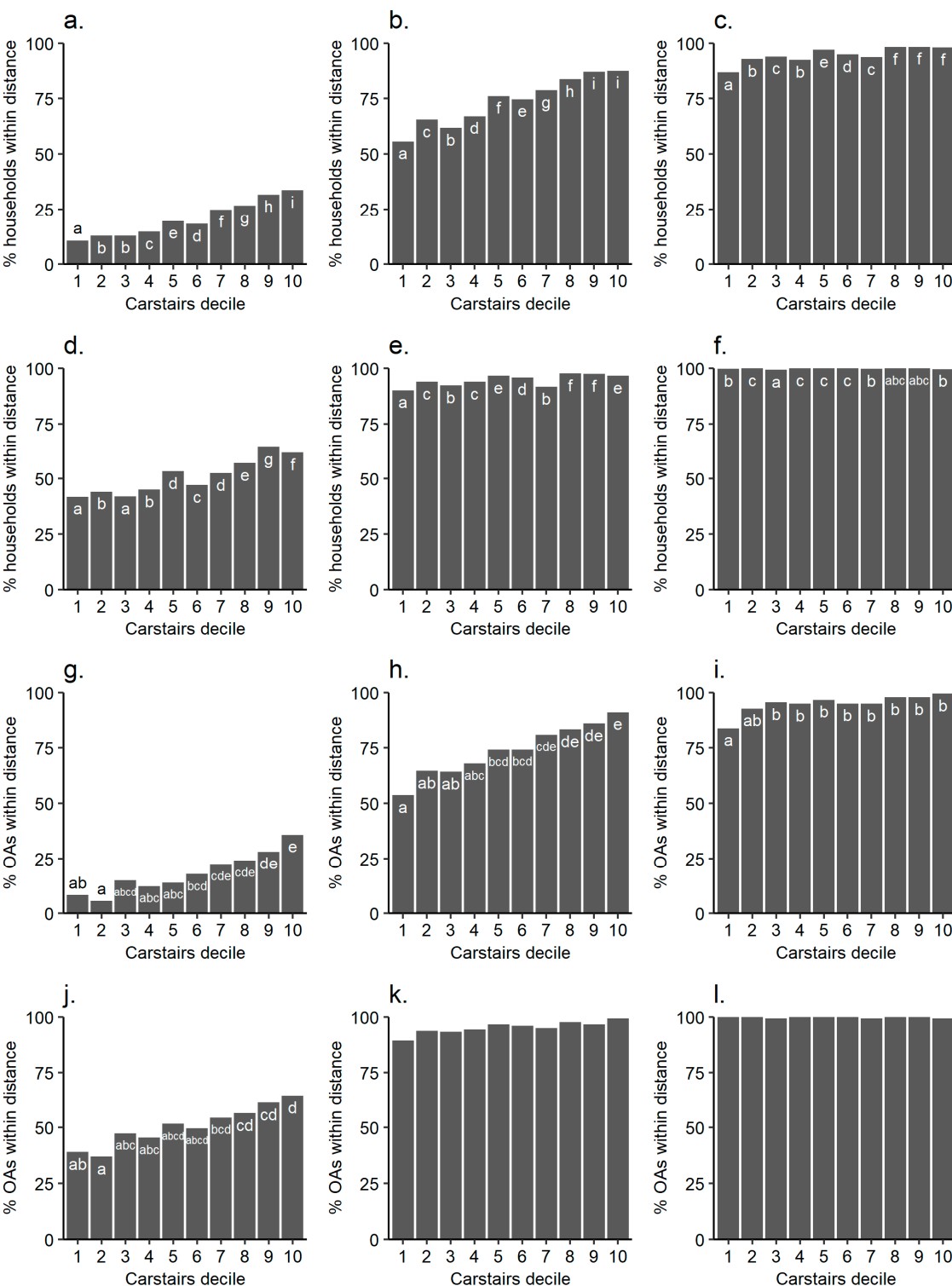

**Figure 3.** Variation in greenspace accessibility by decile of Carstairs Deprivation Index. Accessibility quantified at household scale by network analysis at 100 m (a), 300 m (b) and 500 m (c) and by straight-line buffer at 100 m (d), 300 m (e) and 500 m (f); and at Output Area scale by network analysis at 100 m (g), 300 m (h) and 500 m (i) and by straight-line buffer at 100 m (j), 300 m (k) and 500 m (l). Different letters indicate significant differences among deciles, e.g., 'a' indicates a decile that is significantly different to 'b' but not different to other deciles marked 'a'; while a decile marked 'ab' is not significantly different to those marked either 'a' or 'b' (Tukey's test at $\alpha = 0.05$; multiple comparisons are shown only where the overall ANOVA $p < 0.05$ and significant differences were found).

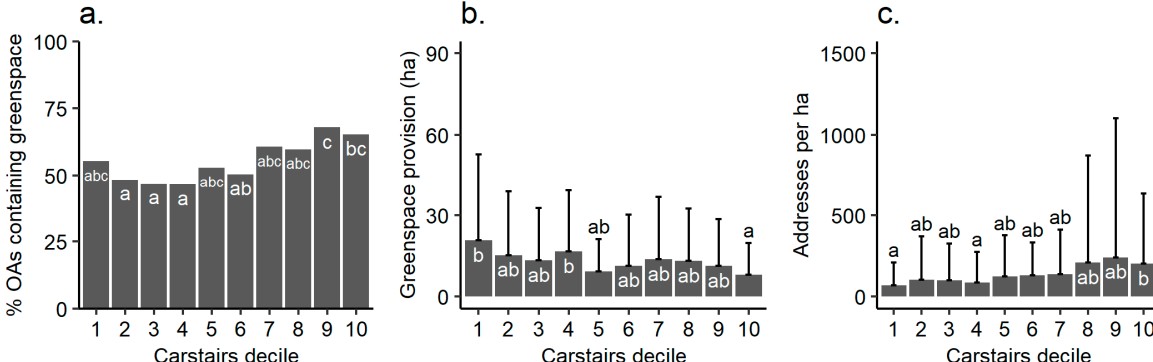

**Figure 4.** Variation in greenspace accessibility (a), provision (b) and population pressure (c) by decile of Carstairs Deprivation Index, quantified at the OA scale according to areal coverage by greenspaces. Positive error bars show one standard deviation. Different letters indicate significant differences among deciles, e.g., 'a' indicates a decile that is significantly different to 'b' but not different to other deciles marked 'a'; while a decile marked 'ab' is not significantly different to those marked either 'a' or 'b' (Tukey's test at $\alpha = 0.05$).

*3.2. Provision*

The results of ANOVAs testing interdecile differences in greenspace provision for households/OAs with access to at least one greenspace are shown in Table 1, with decile means shown in Figure 5 and for OA-scale areal coverage in Figure 4b. The proportions of deviation explained by Carstairs decile are even lower than for accessibility, with only one model explaining more than 5%. Nevertheless, only one model is not statistically significant, and significant multiple differences were found in all except two. It should be noted that in some cases, multiple differences do not follow linear trends with decile means (e.g., groups c and d in Figure 5b); this is because estimated marginal means are calculated accounting for differences in sample sizes for each decile, which due to variation in the proportion of households/OAs with access varies considerably.

In contrast with accessibility, there is no clear trend in provision with deprivation levels, although in the 100 m household scale models and areal coverage model the four deciles of lowest deprivation have slightly greater provision than other deciles (Figure 5a,d, Figure 4b). Significant differences are minimal in the OA scale models due to smaller sample sizes than for household scale combined with small differences in means.

Similarly to accessibility, analysing at household versus OA scale makes little difference to overall estimates of provision (compare Figure 5a–f with Figure 5g–l), except at 100 m, where the household scale suggests 38% greater provision using network analysis, or 8% greater using buffers. Buffer analysis again results in overestimation (compare Figure 5d–f with Figure 5a–c, and Figure 5j–l with Figure 5g–i). At the household scale, the degree of overestimation increases with distance (139%, 167% and 175% of the network analysis values at 100 m, 300 m, and 500 m respectively), but at OA scale this is constant (between 173–178% at all distances).

The results of the areal coverage analysis are most similar to network and buffer analyses at 300 m (Figure 4b compared to Figure 5b,e,h,k).

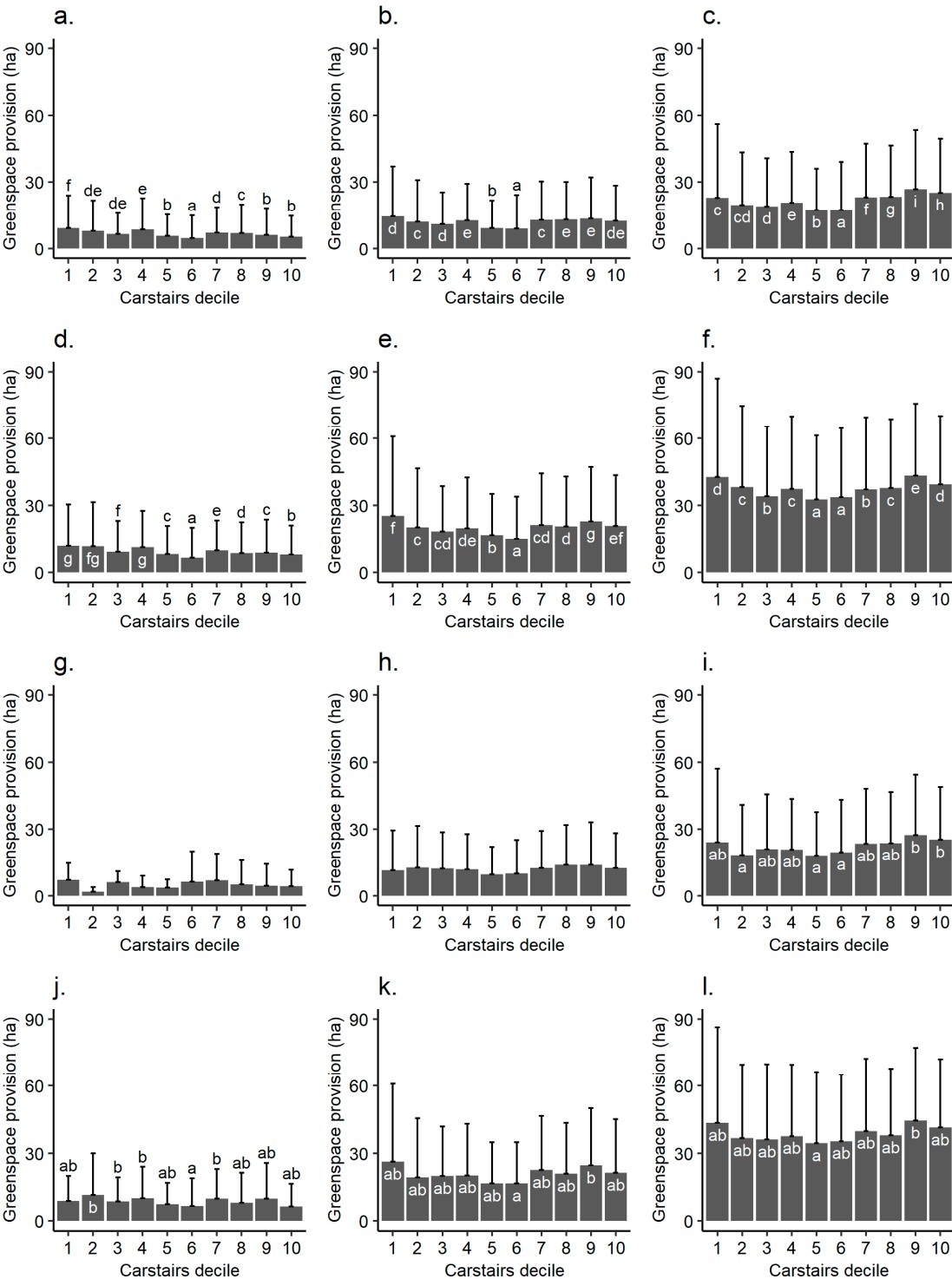

**Figure 5.** Variation in greenspace provision by decile of Carstairs Deprivation Index, for areas with access to at least one greenspace. Provision quantified at household scale by network analysis at 100 m (a), 300 m (b) and 500 m (c) and by straight-line buffer at 100 m (d), 300 m (e) and 500 m (f); and at Output Area scale by network analysis at 100 m (g), 300 m (h) and 500 m (i) and by straight-line buffer at 100 m (j), 300 m (k) and 500 m (l). Positive error bars show one standard deviation. Different letters indicate significant differences among deciles, e.g., 'a' indicates a decile that is significantly different to 'b' but not different to other deciles marked 'a'; while a decile marked 'ab' is not significantly different to those marked either 'a' or 'b' (Tukey's test at $\alpha = 0.05$; multiple comparisons are shown only where the overall ANOVA $p < 0.05$ and significant differences were found).

*3.3. Population Pressure*

Table 1 shows the results of ANOVAs analysing differences in greenspace population pressure. Group means and multiple differences are shown in Figures 6 and 4c. The proportions of deviance explained by decile of Carstairs deprivation are on average intermediate to those for accessibility and provision, with a mean of 4%. All of the models are statistically significant and contain significant multiple differences. This is despite decile standard deviations being larger than the means, in some cases substantially so (Figure 6e,f,j,k).

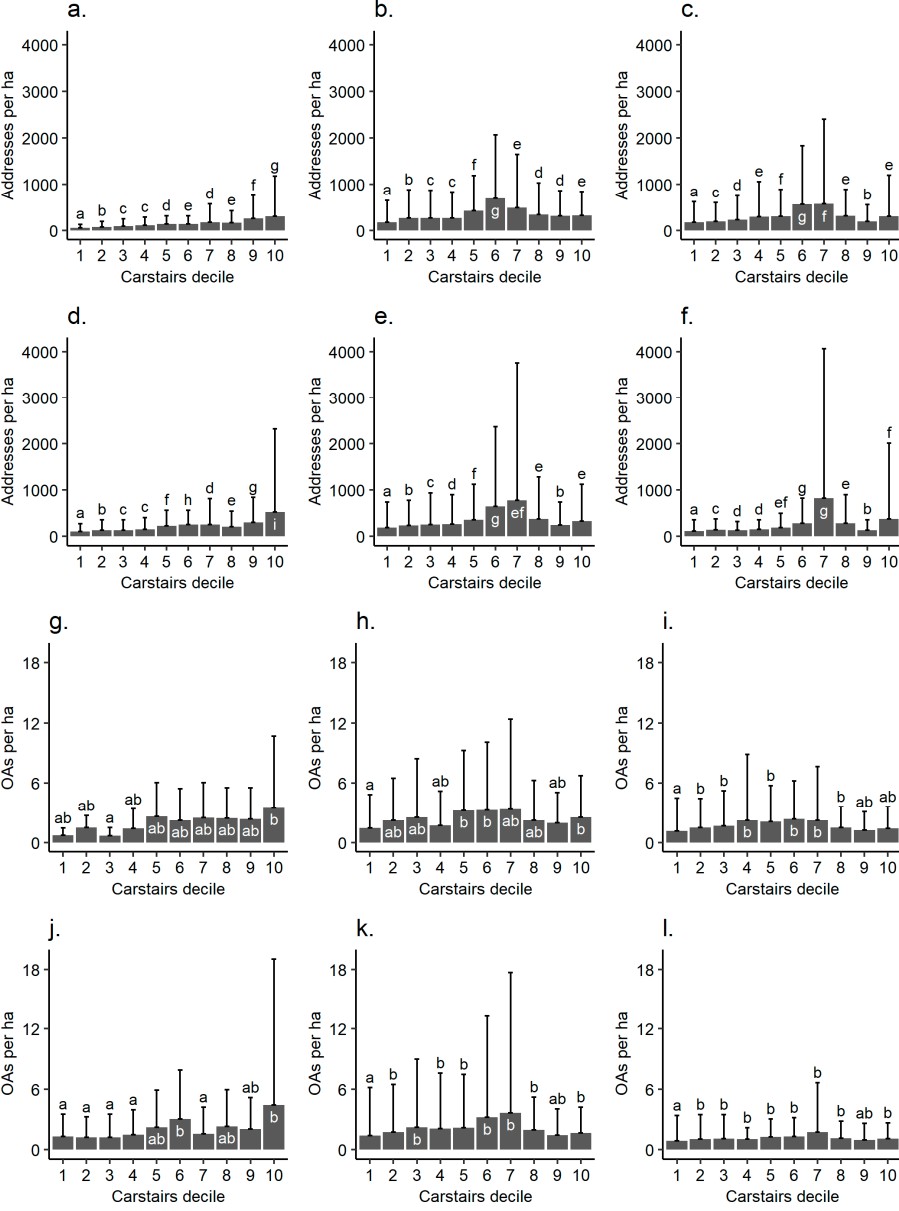

**Figure 6.** Variation in greenspace population pressure by decile of Carstairs Deprivation Index, for areas with access to at least one greenspace. Population pressure quantified at household scale by network analysis at 100 m (a), 300 m (b) and 500m (c) and by straight-line buffer at 100 m (d), 300 m (e) and 500 m (f); and at Output Area scale by network analysis at 100 m (g), 300 m (h) and 500 m (i) and by straight-line buffer at 100 m (j), 300 m (k) and 500 m (l). Positive error bars show one standard deviation. Different letters indicate significant differences among deciles, e.g., 'a' indicates a decile that is significantly different to 'b' but not different to other deciles marked 'a'; while a decile marked 'ab' is not significantly different to those marked either 'a' or 'b' (Tukey's test at α = 0.05).

Relationships between deprivation decile and population pressure vary by distance. For the household scale at 100 m, there is a general association between increasing deprivation and increasing population pressure, i.e., greater greenspace crowding in more deprived areas (Figure 6a,d). At 300 m, the population pressure first increases, then decreases with deprivation (Figure 6b,e). Increasing again to 500 m, there is still a peak at intermediate levels of deprivation, with a possible second peak at very high deprivation (Figure 6c,f).

At OA scale, there are few statistically significant differences due to small differences in means and small sample size (compared to household-scale). The Figures hint at the same distance-dependent patterns as at household scale: at 100 m, in general, the four deciles of lowest deprivation have lower population pressure, and the most deprived decile has the greatest population pressure (Figure 6g,j); and there are indications of a peak at intermediate levels of deprivation at 300 m and 500 m (Figure 6h,i,k,l). However, these observations are not strongly supported by the multiple difference tests. The areal coverage measure shows a similar pattern to the 100 m OA scale analysis, but again is very weak (Figure 4c).

In contrast to the other measures, using straight-line buffers does not consistently produce higher estimates than network analysis. For the household scale, the overall estimate using buffers is 152%, 103% and 83% of that using network analysis at 100 m, 300 m and 500 m, respectively. For OA scale, the equivalent figures are 104%, 88% and 66%.

## 4. Discussion

The objective of this study was to undertake a comparison of answers to the question of how local greenspace availability varies with socioeconomic deprivation when alternative methodological decisions are made. The answers vary depending on the component of availability measured (accessibility, provision or population pressure; Figure 3, Figure 5, Figure 6), and on both the method used to construct local neighbourhoods (straight-line versus network buffers) and neighbourhood size (100 m, 300 m, 500 m). In contrast, using individual households versus OA population centroids makes little difference; although using whole OAs as neighbourhoods does give somewhat different results (Figure 4).

The influence of both neighbourhood size and method of neighbourhood construction is particularly clear. When looking at accessibility, there is a clear relationship between deprivation and accessibility at 100 m, and also at 300 m using the network buffers (Figure 3). However, for larger neighbourhoods, this relationship is not apparent. This is because, due to the relatively high density of greenspaces in the populated parts of Sheffield (Figure 2), nearly all households and OA population centroids have access at 500 m.

This highlights the importance of selecting a suitable neighbourhood size for the local context. Another UK study from Bristol found that 25% of a geographically stratified sample of households were less than 100 m by network distance from their nearest greenspace (compared to 21% in this study), while 75% were within 500 m (compared to 95%), indicating a slightly wider distribution of distances than in Sheffield but with a broadly similar average [34]. Indeed, the average distance to greenspaces in European urban areas appears to be relatively low, and thus a neighbourhood size similar to those used in this study is most appropriate for detection of variation. A study looking at Porta, Portugal found that 80% of neighbourhoods were within 800 m of a greenspace [33], and a study of German cities also found low average distances [29]. However, the same may not be true of US cities. Studies of Baltimore, Maryland and Los Angeles, California, found that only 26% and 29%, respectively, of residential properties were within 400 m by straight-line buffer of a park [13,15]. Although these studies both detected socioeconomic disparities, there is a risk that using too small a neighbourhood may mask variation, just as using too large a neighbourhood (500 m) did in this study.

The effect of neighbourhood size on provision is also clear, and is expected: the larger the area considered, the more greenspace exists within it (Figure 5). It is, however, noteworthy that provision does not increase linearly with the area included in buffers (which would scale approximately with the square of the distance). It is not clear why this should be, although such a pattern could arise if

greenspace were concentrated close to centres of population. For example, if population centres were surrounded mostly by other land uses such as industry or agriculture that do not require the type of greenspace infrastructure desirable in residential areas, expanding the neighbourhood further would not capture much additional greenspace.

The effect of neighbourhood size on both accessibility and provision also highlights the need to put care into considering the true geographic context [22,23]. If the average resident experiences all areas within 500 m of their home on a regular basis, then the fact that all households have access to a greenspace within 500 m would be an important and interesting finding. However, if residents are not prepared to travel 500 m, then the suggestion that everyone has access to a greenspace would be misleading. This would also be the case if residents were not able to travel to all areas within 500 m, for example, if large roads present a physical barrier to crossing, or if a neighbourhood perceived as less safe presents a social barrier. The inclusion of such barriers in network analysis is technically possible, but mapping all barriers across a city is likely to be prohibitively labour-intensive. Using GPS tracking to map routes that are actually travelled may be an alternative, more automated approach to identifying barriers as well as distances travelled.

In contrast, neighbourhood size has little effect on population pressure, which is an area-standardised measure (Figure 6). This pattern has also been observed in a study of the Island of Montreal: the area-proportional greenspace changes little with increases in neighbourhood size, meaning that while the total area of greenspace provided increases, its population pressure does not [37]. These measures are therefore perhaps less sensitive to choice of neighbourhood size, although the choice would still need to reflect a scale relevant to residents' day to day experiences [22]. They are also less sensitive to use of straight-line versus network buffers; although due to the fact that straight-line buffers are always larger than network buffers, the total estimated provision is larger. Other authors have also recommended the use of network buffers to minimise overestimation of provision and accessibility [8,58,59].

Surprisingly, the use of individual households versus OA population centroids has little effect (compare a–f with g–l of Figure 3, Figure 5, Figure 6). It is usually assumed that aggregation results in a loss of information and therefore ability to detect patterns is reduced, i.e., ecological bias is caused [20,60]. This may be due to the fact that urban OAs are of relatively small size (median area of 5.4 ha, which if OAs were circular would be equivalent to a radius of 74 m): bias is more likely where data are aggregated across larger spatial units [20]. Indeed, preliminary work for this study indicated that Lower-layer Super Output Areas, a larger aggregation of English census data with an average population of 1614 (compared to 309 for OAs) [17], showed much weaker patterns than those for OAs or households (LSOAs were not included in the final analysis due to poor statistical power arising from small samples; there are only 345 LSOAs in Sheffield). The main advantage of using individual households has been statistical power. The decile counts for OAs are in some cases quite low for the provision and population pressure measures, as these analyses include only OAs with access to at least one greenspace (Section 2.4). However, visual inspection suggests that there are no obvious patterns that are not statistically significant in the multiple comparison tests at OA scale.

By comparison, the results of the areal coverage analysis of accessibility show a substantially different pattern to that produced by buffer analyses (Figure 4a), indicating that the choice of neighbourhood construction can lead to bias. The trend between deprivation levels and accessibility is considerably flattened, indicating a loss of information. Areal coverage patterns of provision and population pressure (Figure 4b,c) are not so starkly different to those from the buffer analyses. This is an important finding that relates to the UGCoP, because the boundaries of census geographies are unlikely to represent the neighbourhood that residents experience on a day-to-day basis. Using census geographies as a spatial unit of analysis may therefore not effectively capture relationships between populations and their environment.

Also of key interest is that interpretations of the relationship between deprivation and greenspace availability vary by whether accessibility, provision or population pressure is quantified. Looking only at accessibility (except at 500 m, where accessibility has saturated), one would conclude that there is

greater accessibility in more deprived areas (Figure 3). However, no clear pattern is observed in the relationship between deprivation and provision (Figure 5), which would suggest that greenspaces in less deprived areas are larger, and therefore more likely to provide health benefits [21]. The patterns between deprivation and population pressure vary with neighbourhood size (Figure 6), but at 100 m one would conclude that the potential for greenspace crowding is greatest in more deprived areas, and at 300 m and 500 m that the crowding potential is greatest in moderately deprived areas. Thus, depending on methodological decisions, one might conclude that greenspace availability favours more deprived areas; that there is no relationship with deprivation; or that the situation is in fact least favourable in deprived areas.

A positive relationship between deprivation and greenspace accessibility but a negative relationship between deprivation and population pressure was also found for Los Angeles, California, and Baltimore, Maryland [13,15]. Wolch et al. [15] suggested this was due to low density housing in less deprived communities resulting in greater distances to greenspaces; while more deprived areas with much higher housing density meant that greenspaces there would be more likely to become crowded, with this issue being exacerbated by youth from low-income backgrounds being more reliant on public parks to meet their social and play needs. The causes are likely to be similar in Sheffield, with additional issues caused by the fact that high quality greenspaces are relatively less well provisioned in areas of greater deprivation in this city [27].

Similarly, a study of five Australian cities found greater accessibility in areas of greater deprivation, but a more complex, and largely opposite, relationship for provision [36]. Heckert [32] reported a similar pattern in Philadelphia, Pennsylvania. It is clear that this study is not unique in finding that different aspects of greenspace availability show divergent relationships with deprivation.

It is interesting in this context to compare this study's results against those of Barbosa et al. [30]. They undertook a network analysis of greenspace accessibility in Sheffield, using household-level data to estimate deprivation as well as calculating the network distance from each household to the nearest greenspace. Despite use of a different method of deprivation estimation, they also found that more deprived groups live closer to a greenspace on average, indicating that the use of aggregated deprivation data in this study has not introduced excessive bias. However, they calculated that only 36.5% of households are within 300 m of a greenspace, compared to 73.4% in this study. Although there are small differences in methodology and urban extent, these are unlikely to have caused the disparity. The most likely explanation is that they used a different approach to identify greenspaces and whilst an effort was made to include "every parcel of land classified as natural surface . . . which [they] judged to be publicly accessible" [30] (p.188), a number of greenspaces, especially smaller ones, appear not to have been identified. This highlights the importance of accurate accessibility data: while data on where greenspaces are increasingly widely available, data on points from which those greenspaces can be accessed from are still relatively lacking.

Even where methods are fully standardised, there can be large differences in results between study areas [18]. For example, cities in Australia show different relationships between greenspace provision and deprivation [38]. These differences carry over to direct effects on health: in one study, an association between more neighbourhood greenspace and better general health was found in Barcelona, Spain for low-educated residents, but not in Kaunas (Lithuania), Doetinchem (The Netherlands) or Stoke-on-Trent (UK) for any demographic group [61]; and while a positive association between the extent of neighbourhood greenspace and birth weight has been very widely replicated, a study from Pennsylvania failed to find a relationship [18].

These differences highlight a need for consideration of local context when looking for relationships between deprivation (or health) and greenspace availability or exposure. For example, in many US cities, the dominant forces may be a history of racially discriminatory planning practices [13]. In contrast, in UK cities the mid-nineteenth century public health policy of establishing parks in working class neighbourhoods may be a driving factor; indeed, this is the reason for the relationship between better greenspace accessibility and greater deprivation observed in this study [27]. There is in general an identified need for more

precise hypotheses of causative processes and consideration of the complexities of both deprivation and greenspaces [18,26].

*Limitations*

Taking these points into consideration, there are some limitations of this study. First, in the household-level analysis, the use of an area-level measure of deprivation may have resulted in some bias, e.g., due to the ecological fallacy, even though it has not caused the results to diverge from those of a related study using purely household-level data [30]. There may also be bias introduced by assessing at household-level, rather than the level of the individual person, due to variation in average headcount per household across the city.

Second, although we have used a network approach to estimating distances between houses and greenspaces, there may be barriers to access that we have not included, such as locked gates or areas perceived as unsafe to walk through. There is also a possibility that we have not identified all access points, especially unofficial accesses such as gaps in fences.

Third, while the purpose of this study was to investigate the consequences of methodological decisions relating to certain aspects of the complexity of the real-world situation (neighbourhood size and construction, aspect of greenspace availability considered), there are many more methodological issues that we have not been able to address. Key amongst these is the explicit assumption that all the greenspaces included are equally appealing and beneficial to visit, and that any greenspaces not included have no effect. Given that greenspace characteristics including size, quality and amenities all contribute to usage preferences [21], this assumption is unlikely to be correct. Similarly, there are a variety of individual-level contributors to patterns of greenspace usage, including socioeconomic factors and constraints, personal values, and cultural perceptions, which are also likely to affect the benefits obtained from visiting greenspaces [33,34,62–64].

Finally, a limitation related to the data we have used is that our datasets related to different points in time (ranging from 2008–2017). One of our datasets (the PPG17 assessment) is also not available at a national scale. While other local authorities have carried out their own PPG17 assessments, these may not be directly comparable to Sheffield's, and may not be available for all areas. PPG17 assessments aim to include all greenspaces contributing to open space, sports and recreation provision. While the OS Greenspace product can be used to identify many such sites, the Open version does not include all accessible greenspaces while the Academic version does not currently include access points. This limits the ability to perform comparable analyses over a wider area, and therefore test whether there are generalisable patterns between deprivation or health and greenspace access, provision and population pressure.

## 5. Conclusions

This study has produced methodological recommendations that will be relevant to future GIS-based investigations of relationships between local greenspace availability and deprivation or health. First, because straight-line distances considerably overestimate potential exposure by failing to consider the actual routes available for people to travel, network-based calculations of distance are preferable. Second, it is important to select a suitable neighbourhood size, not only in the context of residents' day-to-day experiences, but also because of the possibility of failing to identify variation in greenspace availability due to either saturation of the measure or failure to include nearest greenspaces. Third, we have demonstrated that small-area aggregations of data do not necessarily result in loss of unacceptable amounts of information, in spite of the possibility of ecological bias and the MAUP. This finding may increase the computational feasibility of estimating exposure for larger study areas. Fourth, it is recommended to test different neighbourhood sizes and aggregation levels to guard against the possibility of introducing bias or failure to detect variation. Finally, a consideration of the local urban context and of the complexities of relationships between deprivation and greenspace

is critical, in order to understand the reasons behind observed patterns and to propose policies to improve greenspace distribution equity.

**Author Contributions:** Conceptualization, Meghann Mears and Paul Brindley; Data curation, Meghann Mears and Paul Brindley; Formal analysis, Meghann Mears; Methodology, Meghann Mears and Paul Brindley; Visualization, Meghann Mears; Writing – original draft, Meghann Mears; Writing – review & editing, Meghann Mears and Paul Brindley.

**Funding:** This research was funded by the Natural Environment Research Council, Biotechnology and Biological Sciences Research Council, Arts and Humanities Research Council & Department for Environment, Food and Rural Affairs grant number [NE/N013565/1].

**Acknowledgments:** We thank Jie Qi, Will Allsworth and Luke Ferriday for their contributions to mapping greenspace access points.
        This work was completed as part of the Valuing Nature Programme, a five year £6.5M research programme which aims to improve understanding of the value of nature both in economic and non-economic terms, and improve the use of these valuations in decision making. It funds interdisciplinary research and builds links between researchers and people who make decisions that affect nature in business, policy-making and in practice. See www.valuing-nature.net.
        Census data were sourced from the Office for National Statistics and are © Crown Copyright 2018. OS AddressBase Plus was supplied by Ordnance Survey.

**Conflicts of Interest:** The authors declare no conflict of interest.

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
