# Peer review of "Measuring Urban Greenspace Distribution Equity: The Importance of Appropriate Methodological Approaches"

_ijgi, doi:10.3390/ijgi8060286_

Round 1

Reviewer 1 Report

This is a nice paper on a hot topic in the scientific world, how can we better assess the distribution of green spaces and the equity on accessing it by the population. I congratulate the authors for their work.

Author Response

Reviewer’s comments are shown in blue text; our comments are shown in black. Note that all line references refer to the revised manuscript with tracked changes turned on. 

This is a nice paper on a hot topic in the scientific world, how can we better assess the distribution of green spaces and the equity on accessing it by the population. I congratulate the authors for their work.

We thank the reviewer for their kind comments.

Reviewer 2 Report

The authors conducted a very interesting study focusing on an important issue in the research area of environmental green space inequalities. Therefore, I highly recommend to publish this paper. However, I have some revision mainly in the method and result section.

Introduction

1.       Why did he authors choose the term equity and not equality? I would highly recommend to give a theoretical introduction for the terms “inequality” and “inequity” and justify why this paper used the term “inequity”. The reference from Whitehead mentioned below provides a good overview for these terms in the context of health.

Whitehead, M. The concepts and principles of equity and health. Int. J. Health Serv. 1992, 22, 429–445, doi:10.2190/986L-LHQ6-2VTE-YRRN.

Statistical analysis

1.       Where does the term “binomial ANOVA” come from? When I went through the paper, especially the result section, from my point of view the authors conducted a one-way ANOVA which compares mean values across categories. It should become clear in the whole paper, especially in the result section, that mean values were compared across categories. Moreover, I recommend to give a deeper methodological introduction into the ANOVA procedure and the post-hoc analysis.

2.       I miss a paragraph in the methods where the statistical pre-conditions of an ANOVA were considered, such as balanced data, testing for homogeneity of variances, and normal distributions. If some of the statistical assumptions are not fulfilled, the authors should use other statistics, such as WELCH-ANOVA or non-parametric methods (Kruskal Wallis test with adequate post-hoc tests, such as the Dwass, Steel. Critchow-Fligner post-hoc test)

Results

1.       The quality and resolution of all figures is unacceptable! It is very difficult to read the figures and indicate the letters within the charts. Please provide figures with a better quality and maybe a bigger size in order to ensure that everything is readable.

2.       The presentation of the post-hoc results from the Tukey test are not comprehensible and could be improved. First of all, it is impossible to read the letters in the single bars. In the legend it is mentioned that different letters indicate significant differences. This needs further clarification from my point of view.  What does it mean if there is no letter in a bar, such as in Figure 3 k. and l.? What does it mean if there is only one letter in a bar? What does it mean if there is more than one letter in one bar?  For example, in the first bar of Figure 3 I., does an “a” belongs automatically to the first bar? If yes does “ab” in the second bar means that the second bar is only significant from bar “a”, the first bar? However, it is obvious that the third bar is different from the first bar. So why does it contain only a “b”?  I highly recommend to revise the figures in order to make the post-hoc tests results more comprehensible and visible. A potential solution could be to provide the post-hoc results in an extra table in order to present all kind of combinations? Moreover, I would highly recommend to give a detailed description in the method section how the post-hoc results are presented.

Author Response

Reviewer’s comments are shown in blue text; our comments are shown in black. Note that all line references refer to the revised manuscript with tracked changes turned on.

The authors conducted a very interesting study focusing on an important issue in the research area of environmental green space inequalities. Therefore, I highly recommend to publish this paper. However, I have some revision mainly in the method and result section.

We thank the reviewer for their kind comments and for the suggestions that have helped to improve the clarity of our paper. 

Introduction

1.       Why did he authors choose the term equity and not equality? I would highly recommend to give a theoretical introduction for the terms “inequality” and “inequity” and justify why this paper used the term “inequity”. The reference from Whitehead mentioned below provides a good overview for these terms in the context of health.

Whitehead, M. The concepts and principles of equity and health. Int. J. Health Serv. 1992, 22, 429–445, doi:10.2190/986L-LHQ6-2VTE-YRRN.

Thank you for the reference, which we have added to our manuscript. Our understanding from the reference is that inequities are avoidable and unjust differences, whereas inequalities are unavoidable differences. As greenspaces are part of the urban infrastructure, which is subject to alterations through planning and development processes – and because of the positive influence of urban greenspace on human health – we consider differences in access to and provision of greenspaces to be avoidable and unfair. Moreover, the term equity is used in current papers addressing social injustices to refer to distribution of more resources to those in greatest need. In the present case, those most in need are the most deprived groups. This is in contrast to equality, whereby all groups receive the same resources. We have added definitions of equity and equality to clarify this in the introduction. Please also see references below for a definition of equity and equality in this context (first reference) and examples of its usage in this context.

Rigolon, A.; Browning, M.; Jennings, V. Inequities in the quality of urban park systems: An environmental justice investigation of cities in the United States. Landsc. Urban Plan. 2018, 178, 156–169 https://doi.org/10.1016/j.landurbplan.2018.05.026.

Boone, C.G.; Buckley, G.L.; Grove, J.M.; Sister, C. Parks and people: An environmental justice inquiry in Baltimore, Maryland. Ann. Assoc. Am. Geogr. 2009, 99, 767–787 https://doi.org/10.1080/00045600903102949.

Wolch, J.; Wilson, J.P.; Fehrenbach, J. Parks and Park Funding in Los Angeles: An Equity-Mapping Analysis. Urban Geogr. 2005, 26, 4–35 https://doi.org/10.2747/0272-3638.26.1.4.

Sister, C.; Wolch, J.; Wilson, J. Got green? Addressing environmental justice in park provision. GeoJournal 2010, 75, 229–248 https://doi.org/10.1007/s10708-009-9303-8.

Wolch, J.R.; Byrne, J.; Newell, J.P. Urban green space, public health, and environmental justice: The challenge of making cities “just green enough.” Landsc. Urban Plan. 2014, 125, 234–244 https://doi.org/10.1016/j.landurbplan.2014.01.017.

Heckert, M. Access and equity in greenspace provision: A comparison of methods to assess the impacts of greening vacant land. Trans. GIS 2013, 17, 808–827 https://doi.org/10.1111/tgis.12000.

Shen, Y.; Sun, F.; Che, Y. Public green spaces and human wellbeing: Mapping the spatial inequity and mismatching status of public green space in the Central City of Shanghai. Urban For. Urban Green. 2017, 27, 59–68 https://doi.org/10.1016/j.ufug.2017.06.018.

Lines 35-39, text added:Given that deprived groups have the greatest need of goods and services that improve health, failure to provide adequate greenspaces for more deprived groups is considered an environmental inequity (as opposed to an inequality, whereby everybody would receive the same resources, regardless of need) [11,12].”

Lines 605-609, references [11] and [12] added.

Statistical analysis

1.       Where does the term “binomial ANOVA” come from? When I went through the paper, especially the result section, from my point of view the authors conducted a one-way ANOVA which compares mean values across categories. It should become clear in the whole paper, especially in the result section, that mean values were compared across categories. Moreover, I recommend to give a deeper methodological introduction into the ANOVA procedure and the post-hoc analysis.

As described in Section 2.6 (Statistical analysis), the analysis of accessibility was a binomial ANOVA whereby each household or OA was represented as a 1 (has access) or 0 (does not have access) and was categorised according to decile of deprivation. The analyses of provision and population pressure then compared group means for those households that did have access (i.e. a hurdle model). We have added additional detail to the text to clarify this.

Lines 237-257 text changed:

Previous text: “Statistical analysis of accessibility comprised a binomial ANOVA testing whether accessibility varied between deciles of deprivation. Post-hoc Tukey multiple comparison tests were performed where indicated.

Provision and population pressure were analysed statistically using two-stage (hurdle) models, which were required due to the sometimes large number of zeroes arising from points not within any service areas. The first stage was therefore identical to the accessibility analysis (binomial ANOVA), while the second included only points within at least one service area to analyse inter-decile variability. The second stage comprised an ANOVA with post-hoc Tukey multiple comparison tests based on estimated marginal means.

Statistical analyses were carried out in R v3.5.1, with the emmeans library used to perform multiple comparison tests [53,54].”

New text: “Statistical analysis of accessibility comprised a binomial ANOVA testing whether accessibility varied between deciles of deprivation. Each household/OA was represented by either a 1 (has access) or a 0 (does not have access). This analysis was carried out in R v3.5.1 by calculating a type-III ANOVA table (function ‘Anova’ in library car) for a generalised linear model (function ‘glm’) using the binomial family [55,56]. Model diagnostics indicated that the model assumptions (residual distributions and homogeneity of variance) were met in all cases. Post-hoc Tukey multiple comparison tests were performed where p-values indicated a significant ANOVA test at α=0.05. Multiple comparison tests were performed using the emmeans library, which uses estimated marginal means to account for unbalanced design, to test for differences between each pair of deciles using a Tukey correction [57].

Provision and population pressure were analysed statistically using two-stage (hurdle) models, which were required due to the sometimes large number of zeroes arising from points not within any service areas. The first stage was therefore identical to the accessibility analysis (binomial ANOVA), while the second included only points within at least one service area to analyse inter-decile variability. The second stage comprised an ANOVA with post-hoc Tukey multiple comparison tests based on estimated marginal means. Second stage ANOVAs were again carried out using type-III ANOVA tables calculated for generalised linear models, but in this case using the gaussian family. Again, model assumptions were met in all cases. Multiple comparison tests used the same method as for binomial ANOVAs.”

Lines 720,721, reference [56] added.

2.       I miss a paragraph in the methods where the statistical pre-conditions of an ANOVA were considered, such as balanced data, testing for homogeneity of variances, and normal distributions. If some of the statistical assumptions are not fulfilled, the authors should use other statistics, such as WELCH-ANOVA or non-parametric methods (Kruskal Wallis test with adequate post-hoc tests, such as the Dwass, Steel. Critchow-Fligner post-hoc test)

We hope that our response to the previous comment also resolves this issue.

Results

1.       The quality and resolution of all figures is unacceptable! It is very difficult to read the figures and indicate the letters within the charts. Please provide figures with a better quality and maybe a bigger size in order to ensure that everything is readable.

We have uploaded the figures in an alternative file format that we hope will retain its quality.

2.       The presentation of the post-hoc results from the Tukey test are not comprehensible and could be improved. First of all, it is impossible to read the letters in the single bars. In the legend it is mentioned that different letters indicate significant differences. This needs further clarification from my point of view.  What does it mean if there is no letter in a bar, such as in Figure 3 k. and l.? What does it mean if there is only one letter in a bar? What does it mean if there is more than one letter in one bar?  For example, in the first bar of Figure 3 I., does an “a” belongs automatically to the first bar? If yes does “ab” in the second bar means that the second bar is only significant from bar “a”, the first bar? However, it is obvious that the third bar is different from the first bar. So why does it contain only a “b”?  I highly recommend to revise the figures in order to make the post-hoc tests results more comprehensible and visible. A potential solution could be to provide the post-hoc results in an extra table in order to present all kind of combinations? Moreover, I would highly recommend to give a detailed description in the method section how the post-hoc results are presented.

We have uploaded the figures in an alternative file format that we hope will retain its quality. We have added clarification of how multiple comparisons results are represented in the legends for Figures 3-6 (as these all show multiple comparisons results). We added details to the legends instead of the methods section so that the figures can be better understood in isolation from the main text; we hope that this is acceptable.

Lines 276-280, text changed: “Different letters indicate significant differences among deciles (Tukey’s test at α = 0.05; shown only where the overall ANOVA p<0.05 and significant differences were found).” changed to “Different letters indicate significant differences among deciles, e.g. ‘a’ indicates a decile that is significantly different to ‘b’ but not different to other deciles marked ‘a’; while a decile marked ‘ab’ is not significantly different to those marked either ‘a’ or ‘b’ (Tukey’s test at α = 0.05; multiple comparisons are shown only where the overall ANOVA p<0.05 and significant differences were found).”

Lines 284-287, text changed: “Different letters indicate significant differences among deciles (Tukey’s test at α = 0.05).” changed to “Different letters indicate significant differences among deciles, e.g. ‘a’ indicates a decile that is significantly different to ‘b’ but not different to other deciles marked ‘a’; while a decile marked ‘ab’ is not significantly different to those marked either ‘a’ or ‘b’ (Tukey’s test at α = 0.05).”

Lines 324-328, text changed: “Different letters indicate significant differences among deciles (Tukey’s test at α = 0.05; shown only where the overall ANOVA p<0.05 and significant differences were found).” changed to “Different letters indicate significant differences among deciles, e.g. ‘a’ indicates a decile that is significantly different to ‘b’ but not different to other deciles marked ‘a’; while a decile marked ‘ab’ is not significantly different to those marked either ‘a’ or ‘b’ (Tukey’s test at α = 0.05; multiple comparisons are shown only where the overall ANOVA p<0.05 and significant differences were found).”

Lines 356-358, text changed: “Different letters indicate significant differences among deciles (Tukey’s test at α = 0.05).” changed to “Different letters indicate significant differences among deciles, e.g. ‘a’ indicates a decile that is significantly different to ‘b’ but not different to other deciles marked ‘a’; while a decile marked ‘ab’ is not significantly different to those marked either ‘a’ or ‘b’ (Tukey’s test at α = 0.05).”

Reviewer 3 Report

The paper makes an interesting contribution and is worth publishing. My specific comments are below:

1.       Figure 2 – the map – is too small to see anything. It should be enlarged (at least three times its current size) so the reader can see all colors well.

2.       All other figures also need an improvement – they should be bigger and less pixelated; the labels, especially on the Y axis, are hard to read.

3.       The unknown geographic context problem is discussed in the Introduction, but should be also revisited in the Discussion section. The last paragraph talks about it, but does not mention this problem explicitly.

4.       Lines 46, 126, 128 and 499 – why include personal observation of one of the authors as a reference? If it was personal observation of a non-author, that would make sense.

5.       Table 1 – “% with access” column should report numbers as percent, in the range 0-100, not as a fraction. Please change.

6.       Table 1 – “Deviance explained” – is reported here on 0-1 scale, but in the text it is reported as percent. Please change.

7.       Line 270 – “There pattern” should be “This pattern”

Author Response

Reviewer’s comments are shown in blue text; our comments are shown in black. Note that all line references refer to the revised manuscript with tracked changes turned on.

The paper makes an interesting contribution and is worth publishing. My specific comments are below:

We thank the reviewer for their kind comments and for the suggestions that have helped to improve our paper. 

1.       Figure 2 – the map – is too small to see anything. It should be enlarged (at least three times its current size) so the reader can see all colors well.

We have uploaded the figures in an alternative file format that we hope will retain its quality.

2.       All other figures also need an improvement – they should be bigger and less pixelated; the labels, especially on the Y axis, are hard to read.

We have uploaded the figures in an alternative file format that we hope will retain its quality.

3.       The unknown geographic context problem is discussed in the Introduction, but should be also revisited in the Discussion section. The last paragraph talks about it, but does not mention this problem explicitly.

We have added some discussion of our findings as they relate to the unknown geographic context problem.

Lines 407-417, text added: “The effect of neighbourhood size on both accessibility and provision also highlights the need to put care into considering the true geographic context [22,23]. If the average resident experiences all areas within 500m of their home on a regular basis, then the fact that all households have access to a greenspace within 500m would be an important and interesting finding. However, if residents are not prepared to travel 500m then the suggestion that everyone has access to a greenspace would be misleading. This would also be the case if residents were not able to travel to all areas within 500m – for example, if large roads present a physical barrier to crossing, or if a neighbourhood perceived as less safe presents a social barrier. The inclusion of such barriers in network analysis is technically possible, but mapping all barriers across a city is likely to be prohibitively labour-intensive. Using GPS tracking to map routes that are actually travelled may be an alternative, more automated approach to identifying barriers as well as distances travelled.”

Lines 446-450, text added: “This is an important finding that relates to the UGCoP, because the boundaries of census geographies are unlikely to represent the neighbourhood that residents experience on a day-to-day basis. Using census geographies as a spatial unit of analysis may therefore not effectively capture relationships between populations and their environment.”

4.       Lines 46, 126, 128 and 499 – why include personal observation of one of the authors as a reference? If it was personal observation of a non-author, that would make sense.

We have removed these references.

Line 50-51, text removed:(M. Mears, personal observation)

Line 135, text removed:(M. Mears, personal observation)

Line 137, text removed:(M. Mears, personal observation)

Line 543, text removed:(P. Brindley, personal observation)”

5.       Table 1 – “% with access” column should report numbers as percent, in the range 0-100, not as a fraction. Please change.

We have changed the table to show percentages.

Table 1 (directly after line 271), text changed to show percentages instead of proportions throughout.

6.       Table 1 – “Deviance explained” – is reported here on 0-1 scale, but in the text it is reported as percent. Please change.

We have changed the table to show percentages.

Line 268, text changed: “Deviance explained shown as a proportion” changed to “Deviance explained shown as a percentage”

Table 1 (directly after line 271), text changed to show percentages instead of proportions throughout.

7.       Line 270 – “There pattern” should be “This pattern”

Thank you; we have corrected our typo.

Line 293, text changed: “There pattern” to “This pattern”

Round 2

Reviewer 2 Report

All revisions were adequately addressed. Congratulations for this interesting study! Unfortunately, I could not see the final figures in the new version of the manuscript as there were no links to them in the pdf file. However, I trust the authors reply that they provided figures with a better quality.

Looking forward for the final version of the paper!

Author Response

We thank the Reviewer for their comments and can confirm that we have provided high quality .png format figures.